# Seed Size-Number Trade-Off Exists in Graminoids but Not in Forbs or Legumes: A Study from 11 Common Species in Alpine Steppe Communities

**DOI:** 10.3390/plants14172730

**Published:** 2025-09-02

**Authors:** Xiaolong Zhou, Ronghua Duan, Jian Long, Haiyan Bu

**Affiliations:** 1College of Ecology and Environment, Xinjiang University, Urumqi 830046, China; zhouxiaolong@xju.edu.cn (X.Z.); drh@stu.xju.edu.cn (R.D.); 2Key Laboratory of Oasis Ecology, Xinjiang University, Urumqi 830046, China; 3College of Ecology, Lanzhou University, Lanzhou 730000, China; bohy@lzu.edu.cn

**Keywords:** alpine steppe, functional group, reproductive resource, seed trait

## Abstract

Seed size and number are two important components of plant reproductive traits. Previous theoretical studies have suggested that resource limitations lead to a strong trade-off between seed size and seed number. However, empirical evidence from natural communities remains scarce. In this study, the relationship between seed size and seed number was tested at the community level and in three functional groups—graminoids, forbs, and legumes—in a natural alpine steppe community in the Tianshan Mountains. The role of limiting resources in reproduction and in determining trade-off patterns was also examined by treating the reproductive biomass and allocation of each species as a resource pool for producing seeds. Our results showed a significant negative relationship between seed size and seed number at the community level, which indicated that a trade-off between seed size and number existed and that the species that produced large seeds produced fewer seeds and vice versa. This trade-off was detected for the graminoid group but not for the forb or legume group, so the trade-off at the community level was determined primarily by graminoid species. Moreover, the graminoid group had lower reproductive biomass and allocation than the forb and legume groups, indicating that the graminoid species were more strictly limited by reproductive resources. Our study provides evidence of a seed size-number trade-off in a natural alpine steppe community, especially among graminoid species, and the important role of reproductive resources in determining the trade-off.

## 1. Introduction

In the long process of natural selection, resource allocation trade-offs between different functions (e.g., competitive ability vs. dispersal ability, vegetative growth vs. reproduction) during the plant life cycle are closely related to life history strategies [1,2,3,4]. The issue of plant life history strategies is a topic of current interest in model studies and has been widely investigated in empirical research [5,6,7,8]. Many famous theories have been proposed in previous studies, including the r–K theory [9,10] and the leaf–height–seed (LHS) scheme [11]. In these theories, seed characteristics, as key functional traits of plants, are closely associated with reproductive strategies [3,12,13] and play crucial roles in determining plant fitness.

Previous studies on reproductive strategies have assessed the balancing mechanisms associated with seed size and seed number [14,15,16,17]. This is because seed size is closely associated with seed dispersal, germination, seedling establishment, and the distribution patterns of plant populations [18,19,20]. Many studies have shown that larger seeds typically have a greater chance of germinating and the plants surviving to maturity than smaller seeds do [21,22] (but except for burial conditions [23]). This is because larger seeds contain more nutrients, which provide the seedling with the energy it needs to grow and develop [24]. However, producing larger seeds is more costly for the plant, as it requires more resources to produce a single large seed than multiple small seeds [25]. Therefore, the production of larger seeds is subject to a trade-off between the benefits of increased seed size and the associated costs [25,26]. In contrast, plants that produce many small seeds have a greater chance of producing at least some offspring that will survive and reproduce [27,28]. This is because producing a large number of seeds increases the chance that at least some of them will find suitable conditions for germination and growth [29]. However, producing a large number of seeds is also costly for the plant, as it requires a significant amount of resources to produce and maintain those small seeds [30,31]. In summary, plants face a trade-off between seed size and seed number, which affects their reproductive success and fitness, as the resources available from the environment are limited.

The relationships between seed size and number for plants depend on several factors, including environmental conditions [32,33], resource competition [8,34], and phenotypic plasticity [35]. Among these factors, the variation in environmental conditions is the most important. For example, plants growing in resource-poor environments may produce abundant small seeds that increase the chances of offspring survival [13], whereas plants growing in resource-rich environments may produce a few large seeds that ensure the survival of high-quality offspring [36]. Additionally, seed traits vary with plant functional groups such as graminoid, forb, and legume. In general, legume species produce a few large seeds [37], but forb species tend to produce many small seeds in alpine grassland [38]. The graminoid species produce several different types of seeds from small and spherical to large and non-spherical [39]. In addition, animals (particularly livestock) could influence the seed dispersal of different species through endozoochory [40,41]. Therefore, both abiotic and biotic factors affect the trade-off patterns between seed size and seed number, but the underlying mechanisms are still not clear [42].

Research on seed size-seed number trade-offs has a long history, and many studies have indicated that this trade-off plays a critical role in promoting successful reproduction and population maintenance in plant species [43,44]. To date, most previous studies have concentrated on theoretical models and specific models [20,22,45], but empirical evidence is still rare. Moreover, previous empirical studies either focused on specific environments or compared seed size and number in sparse species communities [5,26,46]. Evidence from multispecies natural communities and different plant functional groups is still scarce [47]. In addition, in general, most researchers have shown that resource limitation plays an important role in determining seed size and number trade-off in theory [17,48], but evidence from an experimental approach to this is still lacking.

In this study, we aimed to examine whether a trade-off between seed size and seed number exists in a multispecies natural alpine steppe community and in functional groups of forbs, graminoids, and legumes. The role of limiting resources in determining trade-off patterns was also tested by treating reproductive biomass and allocation as a resource pool that a plant can distribute to the reproductive part [49,50,51]. Specifically, three closely related questions were asked:

Does a trade-off between seed size and number exist in a multispecies natural alpine steppe community?Are the trade-off patterns different in different functional groups?What is the role of limiting resources in determining seed size and number trade-offs?

## 2. Results

Both seed size (mg) and seed number per ramet varied considerably among the 11 common species at our study site. *Stipa purpurea* (2.64 ± 0.24 mg in 2018, 2.72 ± 1.09 mg in 2019) and *Astragalus multicaulis* (1.13 ± 1.30 mg in 2018, 3.74 ± 1.60 mg in 2019) had greater masses than the other species. The two grass species *Festuca ovina* (1772 ± 303.19 in 2018, 858 ± 208.19 in 2019) and *Koeleria cristata* (539 ± 103.12 in 2018, 2301 ± 1055.70 in 2019) produced more seeds per ramet than the other species (Table 1, Figure 1). Although the seed size and number of seeds of the species differed between 2018 and 2019, the ranks of the species were similar in these two years (Figure 1).

A significant negative linear relationship existed between seed size and seed number at the community level in 2018 and 2019, which indicated a trade-off between seed number and seed size in the alpine steppe community (Figure 2a,c). In 2018, the negative relationship between seed number and seed size was not very strong (Figure 2a), with an R^2^ value of 0.092, although the relationship was statistically significant (*p* = 0.0011). In 2019, the relationship between seed number and seed size was strong (Figure 2c), with a high R^2^ and low *p* value (R^2^ = 0.47, *p* < 0.001). At the functional level, for graminoids, there were significant negative relationships between seed number and seed size in both 2018 and 2019 (Figure 2b,d). In contrast, for forbs and legumes, the relationships between seed number and seed size were not significant (Figure 2b,d). In line with these results, significant negative relationships also appeared between the mean value of seed size and number (Appendix A). After the phylogenetically independent contrasts (PICs), the negative linear relationships were still significant in 2018 (R^2^ = 0.34, *p* < 0.05) and marginally significant in 2019 (R^2^ = 0.45, *p* = 0.06), which indicated that the phylogenetic effect did not change the trade-off relationships (Appendix A). After the data for the 2 years were pooled, the relationships between seed size and number were consistent with each year (Appendix A).

The ANOVA and HSD test results showed that the reproductive biomass and reproductive allocation for graminoids were significantly lower than those for forbs and legumes in both 2018 and 2019 (Figure 3). In 2018, the mean reproductive biomasses for graminoids, forbs, and legumes were 0.097 ± 0.08 g, 0.165 ± 0.14 g, and 0.178 ± 0.16 g, respectively. Correspondingly, the mean reproductive biomasses for graminoids, forbs, and legumes were 0.164 ± 0.13 g, 0.259 ± 0.19 g, and 0.306 ± 0.29 g in 2019.

The seed traits varied among the three functional groups (Figure 4), which suggested that the adaptive strategies of the three functional groups may differ. The legume species had the greatest seed size and highest seed nitrogen content, but the lowest seed number (Figure 4a,b,g), whereas the forb species had the lowest seed size and lowest seed nitrogen and phosphorus content (Figure 4b,g,h). Seed number and length of graminoids were higher than those in legumes and forbs, while the seed chemical traits were between the two groups (Figure 4a,f–h). The PCA results indicated that the seed traits of forbs and legumes were distinct, but those of graminoids were not separated from forbs and legumes (Appendix A). PC1 reflects the seed size-number trade-off, with seed number loading positively and seed size-related traits (size, width, and height) loading negatively (Appendix A; Appendix A). In contrast, PC2 represents the capacity of seeds to persist in the soil seed bank, with soil carbon content loading positively and seed length loading negatively (Appendix A; Appendix A). PC1 explained 39.66% of the variance, while PC2 accounted for 21.35% of the variance (Appendix A).

## 3. Discussion

In a two-year field experiment, our results revealed that a trade-off between seed size and number existed in a multispecies alpine steppe community (Figure 2a,c). However, the relationships between seed size and number were inconsistent across the different functional groups, as a trade-off existed in the graminoid group but not in the forb or legume groups (Figure 2b,d). Thus, the seed size-number trade-off at the community level was determined primarily by graminoid species due to their low reproductive biomass and allocation, which indicated that resource limitations were present for graminoid species but not for forb or legume species (Figure 3). In this study, we tested the seed size-number trade-off in multispecies communities and different functional groups and documented that resource limitations play an important role in determining trade-off patterns. Our results were robust across different years and after the effects of phylogenetic relationships among species were eliminated.

The trade-off between seed size and number varies across different plant species, and ecologists have conducted extensive theoretical research on this relationship [49,50]. For example, life history strategy theory suggests that different species evolve different life history strategies (i.e., trade-offs between survival, growth, and reproduction) to maximize survival and reproductive success, including the r-strategy and K-strategy [9,10]. Other studies have explored the effects of resource availability on plants in different growing environments, which is called resource availability theory [45,51,52]. Ben-Hur and Kadmon (2015) explained the coexistence of competing species in terms of a trade-off between competitive ability and colonization ability on the basis of competition-colonization trade-off models [53]. Germain et al. discovered through the cultivation of populations of each species in both humid and arid environments that there is a strong trade-off between seed size and number among species but no consistent trade-off within species, highlighting the importance of the maternal environment in terms of ecological dynamics, particularly in the context of multispecies coexistence [31,54]. Despite the plethora of theoretical studies and a handful of experimental evidence on seed size-number trade-offs, research has been based on either a particular model or a controlled experiment, and studies of natural communities with multiple species or in different functional groups are still lacking. In this study, we provide evidence for this theory in a natural alpine steppe community involving 11 species, especially among graminoid species.

Seed size–number trade-off was detected in graminoids but not in forbs or legumes (Figure 2). As the resource allocation theory predicts, different environments will select for different optimal patterns of resource allocation to reproductive versus vegetative functions [31,55] and different trade-offs in optimal seed size [48,56] with fecundity [57]. Our results strongly support this idea (Figure 2). Moreover, the strategies used by plants to adapt to the alpine environment differed among the three functional groups. At our study site, legume species had the lowest seed number; in contrast, legume species had the greatest seed size and largest seed volume (Figure 3). The seed nitrogen content of legumes is significantly greater than that of forbs and graminoids (Figure 3), which may be caused by the nitrogen-fixing ability of legume species and the high level of nitrogen in their tissues [58]. Thus, legume species tend to exhibit competitive strategies because high seed size and large seed volume are often closely related to a high survival percentage of seedlings [37], and the high seed nitrogen content also indicates that their seeds contain more nutrients for seedling survival and growth. In contrast, forb species produced small, round seeds with the lowest seed size and seed length, while the seed carbon content in the forb group was slightly greater than that in the graminoid and legume groups (Figure 3). These characteristics indicate that forb species prioritize tolerance strategies because small and high-carbon-content seeds might be buried more deeply and persist longer in the soil than large and non-spherical seeds [38]. This conclusion is further supported by our PCA results (Appendix A; Appendix A). In our PCA analysis, PC2 represents the capacity of seeds to persist in the soil seed bank. For forb species, seed carbon content loaded positively (indicating high carbon content), and seed length loaded negatively (indicating round seeds). Moreover, the graminoid group, which had intermediate seed sizes and nutrient contents, has a compromise strategy between competition and tolerance. In fact, the five graminoid species produced two types of seeds: *Stipa purpurea* and *Agropyron cristatum* produced large, non-spherical seeds, whereas *Poa crymophila*, *Koeleria cristata,* and *Festuca ovina* produced small, spherical seeds. The compromise strategy was well adapted to alpine conditions; thus, more graminoid species coexisted at our study site than forb and legume species.

Resource limitations played a crucial role in determining the trade-off between seed size and number at our study site. This refers to the fact that plants have finite resources available to allocate towards seed production, and they distribute these resources effectively based on the environment [22,59]. In this study, we used reproductive biomass and allocation to indicate the resource pool that is available for allocation to seeds in each species. Our results revealed that the reproductive biomass and allocation of graminoids were significantly lower than those of forbs and legumes in 2018 and 2019, which suggests that the reproductive resource pool of graminoids was lower than that of forbs and that legume and graminoid species experienced stricter resource limitations in terms of reproduction. A probable explanation is that the graminoid species undergo both sexual and clonal reproduction at our study site, but the forb and legume species only carry out sexual reproduction (Table 1). Unlike forb and legume species, graminoid species do not need to allocate vast resources to their reproduction because they are also able to produce offspring through tillering. As previous studies have shown, the two reproductive modes, i.e., sexual and asexual, affect resource allocation patterns and the seed size-number trade-off in perennial plants [60]. Species with asexual propagation can establish local populations via rapid, short-distance dispersal and forage in high-quality environments by escaping from poor sites and proliferating at richer sites [61,62]; thus, sexual reproduction enhances the competitive ability of these species, especially under nutrient enrichment conditions. In contrast, sexually reproducing plants are able to disperse seeds long distances, easily colonize new environments, and tolerate more severe habitats [63,64].

At our study site, the soil type is chestnut soil, which contains a high percentage of humus (19.8%) and is a fertile soil. The precipitation was 409 mm and 469 mm in 2018 and 2019, respectively, which was greater than the mean annual precipitation (265.7 mm) in recent decades. In alpine steppe ecosystems, precipitation is a crucial limiting factor. Under conditions of higher precipitation, graminoids may exhibit a greater propensity for asexual reproduction, consequently allocating fewer resources to sexual reproduction (the reproductive allocation is 16.26% and 9.98% in 2018 and 2019, respectively, Figure 3) [65]. This constraint on reproductive resources led to a more pronounced trade-off between seed size and number (Figure 2). Additionally, previous studies have suggested that low temperatures were also a key limiting resource in alpine plant communities [54,66,67,68]. Specifically, previous studies have documented that water (the mean annual precipitation is only 265.7 mm) and nitrogen limited the above-ground biomass in our study site [69]. In the future, more studies are needed to distinguish the important roles of these resources in determining the patterns of seed size-number trade-offs.

In this study, we demonstrated the existence of trade-offs between seed size and number in a multispecies alpine steppe community, primarily driven by graminoid species. The seed size-number trade-off was determined by the reproductive resources and mode of reproduction of the plant. In addition, species in different functional groups may adopt different strategies to adapt to alpine habitats during the seed stage. In the future, more comprehensive studies are needed to test the seed size-number trade-off pattern in other ecosystems and explore the underlying mechanisms in natural communities. Additionally, the role of the seed size-number trade-off in affecting community composition should be investigated because both seed size and seed number are closely related to species fitness in the plant community.

## 4. Materials and Methods

### 4.1. Study Site

This study was conducted in Bayanbulak grassland, near the Bayanbulak Grassland Ecosystem Research Station (42°52′ N, 83°42′ E), which is located in the southern Tianshan Mountains within Hejing County, Xinjiang Uygur Autonomous Region, China. This region is one of the largest stock-breeding bases in Xinjiang and is recognized as a biodiversity hotspot in Central Asia. At our study site, the altitude is approximately 2470 m. On the basis of meteorological data from recent decades (2014–2023), mean annual precipitation is 265.7 mm (mostly concentrated from May to August), mean annual temperature is −4.8 °C, the amount of evaporation is 1022.9–1247.5 mm, total annual sunshine duration is 2466–2616 h, and the annual snow cover duration is 150–180 days. There is no absolute frost-free period, and the climate is typically alpine. The vegetation is alpine steppe, and the dominant plant species are *Stipa purpurea*, *Festuca ovina*, *Agropyron cristatum* (Poaceae), and *Astragalus polycladus* (Leguminosae). Average above-ground dry biomass is 71–382 g/m^2^, and richness is 9–13 species/m^2^.

### 4.2. Experimental Design and Seed Trait Measurements

In 2018 and 2019, from the middle of August to the end of September, 11 common species were sampled from an enclosed and flat alpine steppe community (approximately 5 hectares). This area has been fenced since 2018, and livestock such as sheep and horses have been forbidden throughout the year, but wild animals, including marmots (*Marmota bobac*) and zokors (*Myospalax* spp.), have been allowed inside. These 11 species accounted for more than 97% of the above-ground biomass and 92% of the cover at our study site. For each species, when the seeds ripened naturally, 12–15 healthy plant individuals (including seeds from each individual) were clipped and placed in an envelope, and each individual was at least 10 m apart in this process.

In the laboratory, after the materials were dried at 50 °C in an oven to a constant weight, we removed the reproductive parts of each individual and then weighed them on an electronic balance to 0.0001 g to represent reproductive biomass [70,71]. The reproductive allocation was defined as the ratio of reproductive biomass to individual biomass of each species [70,71]. Then, the number of seeds was counted, and the mass of the seeds in each reproductive part was weighed. In our study, we defined seed size as the seed mass per capita, that is, the total seed mass divided by the number of seeds for each individual. We then measured the six other seed traits, which included three morphological traits (seed length, width, and height) and three chemical traits (seed carbon, nitrogen, and phosphorus contents), of each species. To measure morphological traits, 10 randomly selected seeds from each species were scanned, and the image was analyzed with Epson Expression 10000XL (Seiko Epson Corporation, Suwa, Japan) and imaging software (Win SEEDLETM 2.0, Regent Instruments Inc.,Québec, QC, Canada) [72]. Next, about 2 g of seeds were crushed with a ball mill and used to measure the carbon, nitrogen, and phosphorus contents. The carbon content was measured via the potassium dichromate external heating method. The nitrogen and phosphorus concentrations were analyzed via a continuous flow injection analyzer (SKALAR, Breda, The Netherlands) [73].

### 4.3. Statistical Analysis

All 11 species in our study were divided into three functional groups: forb, graminoid, and legume (Table 1). First, the seed number and size data were log-transformed prior to analysis. Next, we described the distribution of seed size and the number of seeds per individual for different plant species. Next, we used simple linear regression to examine the relationship between seed size and seed number at the community level and among the three functional groups for each year (2018 and 2019) and two years data pooled, respectively. Then, the means of seed size and seed number for each species were calculated, and simple linear regression was used to detect the relationship between them. In this way, we eliminated the random effects of different species. A phylogenetically independent contrasts (PICs) analysis was subsequently performed to test the potential effects of the phylogenetic relationships among species on the correlations between seed size and seed number [74]. To perform this analysis, a phylogenetic tree that included the 11 species was constructed (Appendix A). We downloaded 11 ITS sequences representing each species from NCBI. All sequences were aligned using MEGA 5, and a maximum likelihood (ML) tree was constructed on the basis of the aligned ITS sequence variations using MEGA version 5.0, with default parameters [75].

Finally, the reproductive allocation of each species was calculated as the reproductive biomass divided by the individual biomass [70,71], and the reproductive biomass and reproductive allocation of the three functional groups were subsequently analyzed via ANOVA and the HSD test. To detect the reproductive strategies of the three functional groups, eight seed traits (number, size, length, width, height, carbon content, nitrogen content, and phosphorus content) were subjected to principal components analysis (PCA).

All analyses were carried out in R version 4.2.2 (R Development Core Team, 2019). The PICs analysis was conducted via the “ape” and “ade4” packages, the HSD test was performed via the “agricolae” package, and the data were cleaned and plotted via the “tidyverse” and “ggplot2” packages, respectively.

## Figures and Tables

**Figure 1 plants-14-02730-f001:**
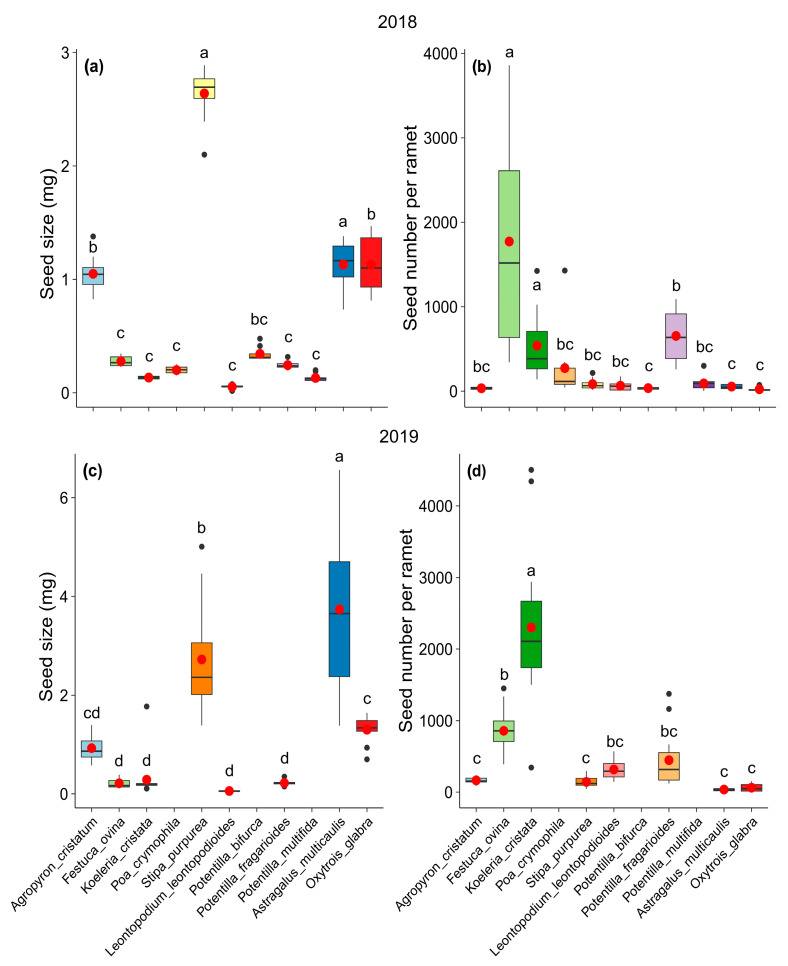
Seed size (**a,c**) and seed number per ramet (**b,d**) of 11 common species at our study site in 2018 and 2019. (Red dots represent the mean value, different lowercase letters indicate statistically significant differences among species (*p* < 0.05)).

**Figure 2 plants-14-02730-f002:**
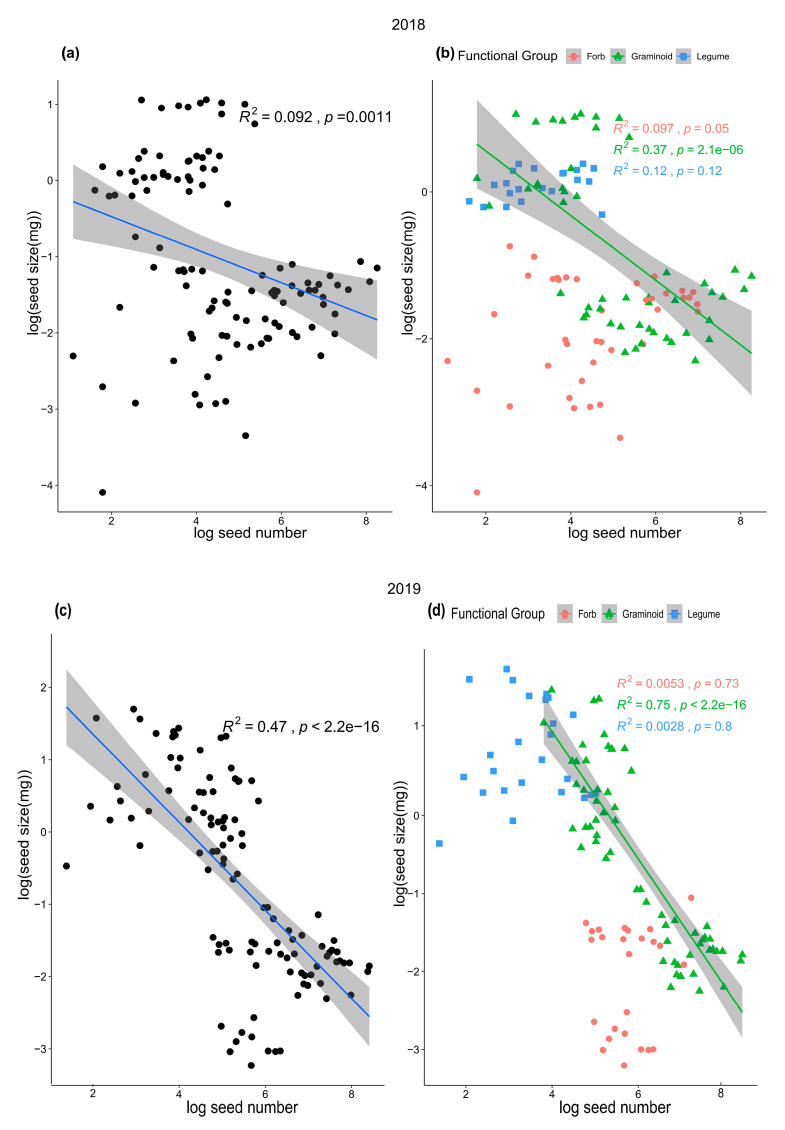
Relationships between seed size and seed number at the community level (**a**,**c**) and at the functional group level (**b**,**d**) in 2018 and 2019. The seed size and seed number data were log transformed.

**Figure 3 plants-14-02730-f003:**
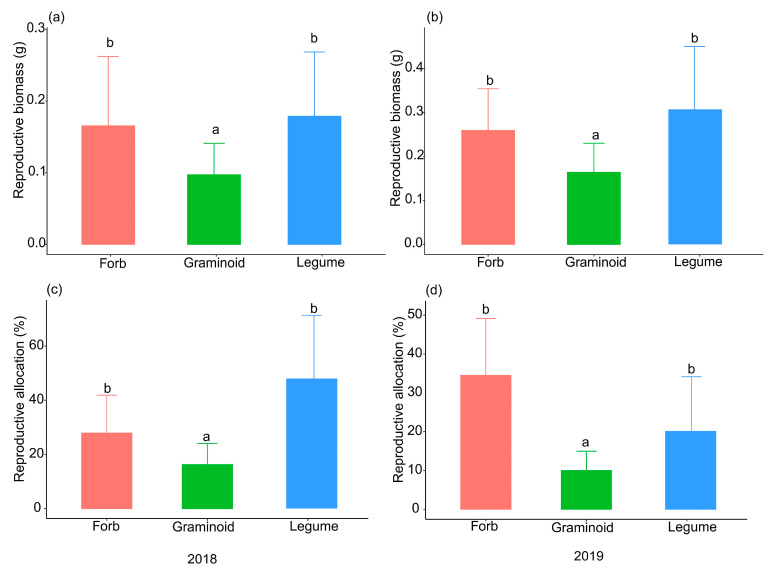
Reproductive biomass and allocation of the three functional groups in 2018 (**a,c**) and 2019 (**b,d**). (Different lowercase letters indicate statistically significant differences among functional groups (*p* < 0.05)).

**Figure 4 plants-14-02730-f004:**
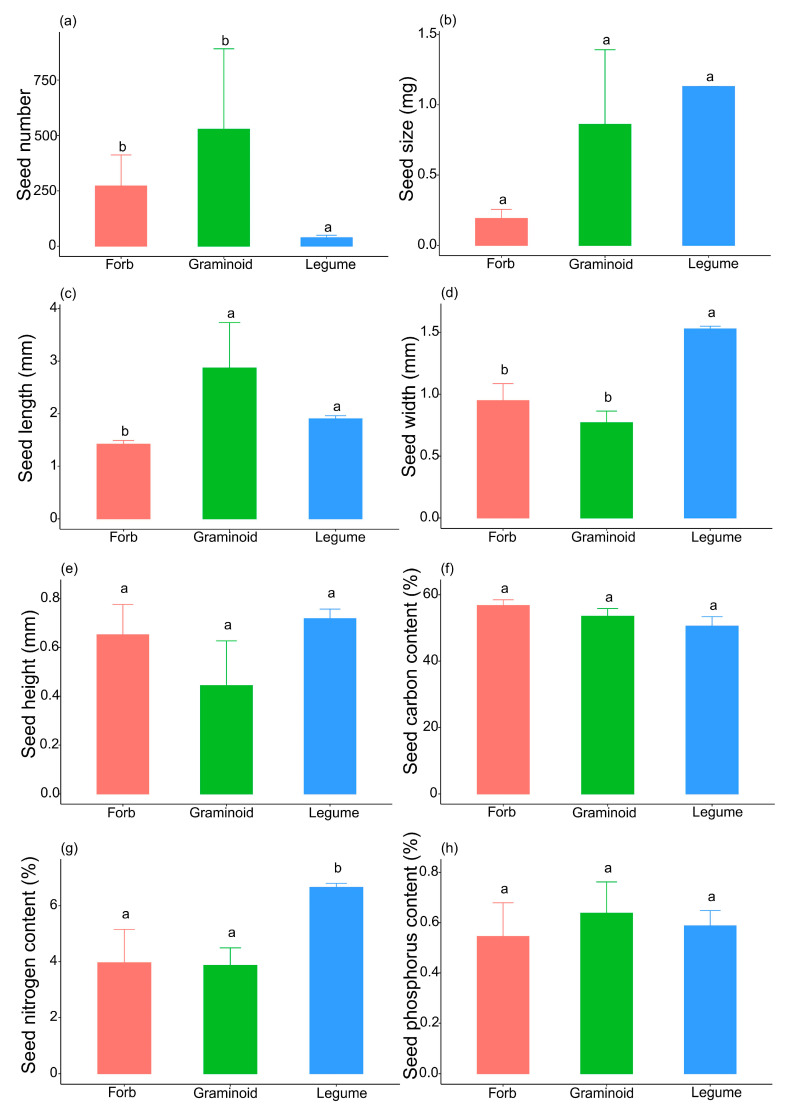
The seed traits of three functional groups. (Different lowercase letters indicate statistically significant differences among functional groups (*p* < 0.05); seed number (**a**), seed size (**b**), seed length (**c**), seed width (**d**), seed height (**e**), seed carbon content (**f**), seed nitrogen content (**g**), seed phosphorus content (**h**)).

**Table 1 plants-14-02730-t001:** The seed size, seed number per ramet, and reproduction types of 11 common species in our study site (the mean value for 2018 and 2019).

Species	Functional Group	Seed Size (mg)	Seed Number	Reproduction Type
*Agropyron cristatum*	graminoid	0.9879 ± 0.16	98 ± 19.82	sexual + clonal
*Festuca ovina*	graminoid	0.2458 ± 0.04	1315 ± 255.69	sexual + clonal
*Koeleria cristata*	graminoid	0.2102 ± 0.02	1420 ± 579.41	sexual + clonal
*Poa crymophila*	graminoid	0.2 ± 0.03	273 ± 193.10	sexual + clonal
*Stipa purpurea*	graminoid	2.68 ± 0.67	113 ± 65.58	sexual + clonal
*Leontopodium leontopodioides*	forb	0.0547 ± 0.02	191 ± 54.92	sexual
*Potentilla bifurca*	forb	0.3431 ± 0.07	36 ± 16.44	sexual
*Potentilla fragarioides*	forb	0.2315 ± 0.03	551 ± 308.05	sexual
*Potentilla multifida*	forb	0.131 ± 0.03	91 ± 82.93	sexual
*Astragalus multicaulis*	legume	2.435 ± 1.3	45 ± 33.91	sexual
*Oxytrois glabra*	legume	1.216 ± 0.25	42 ± 23.05	sexual

## Data Availability

All the required data are uploaded as Appendix A.

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
