# Peer review of "Seed Size-Number Trade-Off Exists in Graminoids but Not in Forbs or Legumes: A Study from 11 Common Species in Alpine Steppe Communities"

_plants, 2025, doi:10.3390/plants14172730_

Round 1

Reviewer 1 Report

Comments and Suggestions for Authors

Title: Seed Size-Number Trade-Off exists in Graminoids but not in Forbs or Legumes: A study from 11 Common Species in Alpine Steppe Communities

The subject of this manuscript falls within the general scope of Plants. The manuscript is an original contribution to seed ecology, improving our knowledge on the trade-off between seed size and number. The manuscript is very well written and organized, including interesting figures, but some recent references should be included. This manuscript should be reconsider after major revision

Keywords should be in alphabetical order. Avoid words included in the title.

Line 46. Here you add this reference, since larger seeds are a bad option for establishment in some conditions such as burial.

https://www.tandfonline.com/doi/abs/10.1080/17550874.2020.1832154

Line 70. You could also add some information about seed size and dispersion. See, for example:

https://www.mdpi.com/2071-1050/12/13/5450

https://onlinelibrary.wiley.com/doi/abs/10.1111/wre.12298

Lines 93-97, Tables A2, A3. Species scientific names must be written in italics.

Lines 93-97. Add standard error to the mean values.

Table 1. Add standard error to the mean values.

Figure 1. You could add letter over the bars to indicate significant differences among species. Indicate in the legend how you are showing the deviations from the means.

Line 129. Add standard error to the mean values.

Figures 3 and 4. Indicate in the legends the meaning of all lowercase letters and the error bars.

Lines 213 and 260. ‘experimental results’. You have NOT carried out an experiment, but a descriptive study.

Line 273. What do you mean exactly with ‘ramet biomass’? Please, explain. Do you mean vegetative biomass? Reproductive biomass is also a part of ramets.

Lines 280-284. Please, add references to support your analyzing methods.

Lines 288 and 302. Did you test the normality and the homogeneity of variance of your data series?

I have not edited the English writing since it is not my native language.

Author Response

The subject of this manuscript falls within the general scope of Plants. The manuscript is an original contribution to seed ecology, improving our knowledge on the trade-off between seed size and number. The manuscript is very well written and organized, including interesting figures, but some recent references should be included. This manuscript should be reconsider after major revision.

Response:Thank you for your valuable comment. Following your suggestions, we have made the following major revisions to the manuscript: (1) We have added eight references to better support our Introduction (line46, 71-72), Discussion (line 247), and Materials and methods sections (line 291-294, 303, 306). (2) We have revised the Figures (Figure 1) and tables (Table A2, A3, A4). (3) We have made revisions throughout the manuscript to improve the overall expression.

Keywords should be in alphabetical order. Avoid words included in the title.

Response:Thank you for this valuable comment. We have re-selected the keywords and arranged them in alphabetical order.

Line 46. Here you add this reference, since larger seeds are a bad option for establishment in some conditions such as burial.

https://www.tandfonline.com/doi/abs/10.1080/17550874.2020.1832154

Response:Thank you for this comment. We have added this reference to the Introduction section (line 45).

Line 70. You could also add some information about seed size and dispersion. See, for example:

https://www.mdpi.com/2071-1050/12/13/5450

https://onlinelibrary.wiley.com/doi/abs/10.1111/wre.12298

Response:Thanks. We have added information about seed size and dispersion and cited two valuable references (line 69-70).

Lines 93-97, Tables A2, A3. Species scientific names must be written in italics.

Response:Thanks. We have revised it according to your suggestions. Please accept our sincerest apologies for our oversight.

Lines 93-97. Add standard error to the mean values.

Response:Thanks. We have revised it according to your suggestions.

Table 1. Add standard error to the mean values.

Response:Thanks. We have added the standard errors.

Figure 1. You could add letter over the bars to indicate significant differences among species. Indicate in the legend how you are showing the deviations from the means.

Response:Thanks for this valuable comment. We have revised Figure 1 according to your suggestions.

Line 129. Add standard error to the mean values.
Response:Thanks. We have added the standard errors.

Figures 3 and 4. Indicate in the legends the meaning of all lowercase letters and the error bars.

Response:Thanks. We have revised the legends of Figure 3 and Figure 4 according to your suggestions.

Lines 213 and 260. ‘experimental results’. You have NOT carried out an experiment, but a descriptive study.

Response:Yes, we have deleted ‘experimental ’ in this sentence. Thanks for your valuable comment.

Line 273. What do you mean exactly with ‘ramet biomass’? Please, explain. Do you mean vegetative biomass? Reproductive biomass is also a part of ramets.

Response:Thanks for your important question. In this study, 'ramet biomass' refers to the plant individual biomass. This terminology was adopted because our research includes both graminoid species that exhibit clonal growth and herb species, thus leading us to use 'ramet' as a unifying term. As you correctly noted, this terminology could lead to reader confusion in this context. Therefore, we have replaced 'ramet biomass' with 'individual biomass' for clarity. You are right, reproductive biomass is a part of ramets. In our study, the reproductive allocation was defined as the ratio of reproductive biomass to individual biomass of each species. This definition has also been employed by previous studies (Niu et al. 2008, Niu et al. 2012), and we have provided relevant references in this section (line 284).

Reference

Niu, K., Y. Luo, P. Choler, and G. Du. 2008. The role of biomass allocation strategy in diversity loss due to fertilization. Basic and Applied Ecology 9:485-493.

Niu, K., B. Schmid, P. Choler, and G. Du. 2012. Relationship between Reproductive Allocation and Relative Abundance among 32 Species of a Tibetan Alpine Meadow: Effects of Fertilization and Grazing. PLoS ONE 7:e35448.

Lines 280-284. Please, add references to support your analyzing methods.

Response:Thanks for this valuable comment. We have supplemented the references to support our experimental methods and data analysis (line292-294, 302, 306, 326).

Lines 288 and 302. Did you test the normality and the homogeneity of variance of your data series?

Response:Thanks for your valuable comment. The Shapiro-Wilk test was conducted on the residuals from both the analysis of variance (ANOVA) and linear regression models, and all residuals were found to be normally distributed.

I have not edited the English writing since it is not my native language.

Response:Thank you for your helpful comments. We have amended the language based on the feedback provided by the other reviewers. To ensure clarity and precision, the manuscript's language has also been refined by a colleague with overseas study experience.

Reviewer 2 Report

Comments and Suggestions for Authors

Line 23. Change “presented” to “had”

 Line 24. Delete “did”

 Line 44. Insert “the plants” before “surviving”

 Line 48. Need to reword “plants must carefully balance”    The way this is worded, it sounds like plants have a brain and can make a decision about what to do.  As noted below, I found this problem in several places, so some rewording is needed.

Line 62.  Change “to” to “that”      Otherwise, you are saying that plants have a brain.  Also, some people would say that “to” is non-Darwinian.

Line 63. Change “to ensure” to “that ensure”  -- same problem as on line 62

Line 64. I suggest you change the wording to “Additionally, seed traits vary with plant functional groups…..”

Line 66. Change “tended” to “tend”

Line 67. Delete “enable to”

Line 68. Delete “seeds” after “non-spherical”

 Line 93.  What is the meaning of “size per grain”    Should this be “Both seed size (mg) and seed number”  ?? 

Line 95. Change to “had greater masses than the other species.”

Lines 94-97.  Put the names of the four species in italics

Line 98. Delete “did” after “other species”

Line 99. Change “changed” to “differed”

 Line 99. Change “those” to “these”

Line 101.  Delete “per grain”  -- you clearly mean “mg” , which you have in the table

Table 1.  Put the names of all species in italics.

Line 114.  Change to  “d). In contrast.   --- when you use a “:” and keep extending the sentence you have a run-on sentence, which is not good

Line 120. Change to “After the data for the 2 years were pooled, the “

Line 127.  Change to “for graminoids”  and change to “for forbs”   ---- that is change “of” to “for”

Line 137-138. Change to “lowest seed size and lowest seed nitrogen and phosphorus content…”

Line 138. Need to start a new sentence.  Change to “h). Seed number and ….”

 Line 139. Delete “the” before “seed”

Line 140. Change “between” to “of”

 Line 141. Insert “those” before “graminoids”

Line 149. Change “may cause by” to “due to”

Line 163.  Delete “decisions”   --- not needed and plants have no brains

Lines 179-180.  I suggest you delete “need to make strategic decisions on how to”    -- plants can not make decisions.    All you need to say is “ and they distribute these resources effectively based on….”

Line 198.  This paragraph is much too long.  Start a new paragraph   -- begin with “At our study site….”

Line 209. Delete “The” before  “seed”

 Line 216. Delete “the” before “legume”

 Line 221.  Change “rate” to “percentage”  The word “rate” means the speed, but you do not seem to be talking about speed of survival.

Line 228. Change “selected” to “has”     --- no brains in plants

Line 234. Delete “did”   -- not needed

Line 237.  “mode of the plant”  this is not clear.  Do you mean  “and the mode of reproduction of the plant”?  Or, do you mean the function group to which the species belongs?

Line 252, 252. Delete “the” before “mean”

Line 254. Delete “the” before “total”

Line 255. Delete “the” before “annual”

Line 258. Delete “the” before “average”

 Line 259. Delete “the” before “richness”

 Line 268.  Not very clear.  I suggest you delete “randomly sampled, then”   

Unless you used a random numbers table to help you select the plants, then you sampled in a haphazard fashion.   Did you also put the seeds in the envelope?  Need to make this clear for the reader.

Line 270. Change “part” to “parts”

 Line 271.  Not clear about the weighing.  Did you weight the material as soon as you were back in the lab, i.e. fresh weight?  Or, did you dry the material for several days and then weight it, i.e. air-dried weight?  Or, did you dry the material in a oven to a constant weight, i.e. oven-dry weight?  Need to tell the reader which one is correct.

 Line 275.  Insert “total” before “seed mass”    Is this what you did? 

Line 276.  Change to “measured six other seed traits, which….”

Line 293.  Remove the “:” and start a new sentence.    This will solve the run-on sentence problem.

Comments on the Quality of English Language

My comments/suggestions will help improve the English and the writing. 

Author Response

Comments and Suggestions for Authors

As non-native English-speaking scholars, we sincerely apologize for the language issues present in our manuscript. We are deeply grateful for your patient, meticulous, and professional comments. Following your suggestions, we have revised the entire manuscript, and the language quality has improved substantially.

Line 23. Change “presented” to “had”

Response:Done.

Line 24. Delete “did”

Response:Done.

Line 44. Insert “the plants” before “surviving”

Response:Done.

Line 48. Need to reword “plants must carefully balance”    The way this is worded, it sounds like plants have a brain and can make a decision about what to do.  As noted below, I found this problem in several places, so some rewording is needed.

Response:Many thanks for this valuable suggestion. We have revised this sentences (line 50-51,57-59).

Line 62.  Change “to” to “that”      Otherwise, you are saying that plants have a brain.  Also, some people would say that “to” is non-Darwinian.

Response:Many thanks for this valuable suggestion. We have revised the sentence.

Line 63. Change “to ensure” to “that ensure”  -- same problem as on line 62

Response:Many thanks. We have revised the sentence.

Line 64. I suggest you change the wording to “Additionally, seed traits vary with plant functional groups…..”

Response:Many thanks. We have revised the sentence.

Line 66. Change “tended” to “tend”

Response:Done.

Line 67. Delete “enable to”

Response:Done.

Line 68. Delete “seeds” after “non-spherical”

Response:Done.

 Line 93.  What is the meaning of “size per grain”    Should this be “Both seed size (mg) and seed number”  ?? 

Response:Yes, we have revised it.

Line 95. Change to “had greater masses than the other species.”

Response:Done.

Lines 94-97.  Put the names of the four species in italics

Response:Done.

Line 98. Delete “did” after “other species”

Response:Done.

Line 99. Change “changed” to “differed”

Response:Done.

 Line 99. Change “those” to “these”

Response:Done.

Line 101.  Delete “per grain”  -- you clearly mean “mg” , which you have in the table

Response:Done.

Table 1.  Put the names of all species in italics.

Response:Done.

Line 114.  Change to  “d). In contrast.   --- when you use a “:” and keep extending the sentence you have a run-on sentence, which is not good

Response:Done.

Line 120. Change to “After the data for the 2 years were pooled, the “

Response:Done.

Line 127.  Change to “for graminoids”  and change to “for forbs”   ---- that is change “of” to “for”

Response:Done.

Line 137-138. Change to “lowest seed size and lowest seed nitrogen and phosphorus content…”

Response:Done.

Line 138. Need to start a new sentence.  Change to “h). Seed number and ….”

Response:Done.

Line 139. Delete “the” before “seed”

Response:Done.

Line 140. Change “between” to “of”

Response:Done.

Line 141. Insert “those” before “graminoids”

Response:Done.

Line 149. Change “may cause by” to “due to”

Response:Done.

Line 163.  Delete “decisions”   --- not needed and plants have no brains

Response:Done.

Lines 179-180.  I suggest you delete “need to make strategic decisions on how to”    -- plants can not make decisions.    All you need to say is “ and they distribute these resources effectively based on….”

Response:Done.

Line 198.  This paragraph is much too long.  Start a new paragraph   -- begin with “At our study site….”

Response:Done.

Line 209. Delete “The” before  “seed”

Response:Done.

 Line 216. Delete “the” before “legume”

Response:Done.

 Line 221.  Change “rate” to “percentage”  The word “rate” means the speed, but you do not seem to be talking about speed of survival.

Response:Done.

Line 228. Change “selected” to “has”     --- no brains in plants

Response:Done.

Line 234. Delete “did”   -- not needed

Response:Done.

Line 237.  “mode of the plant”  this is not clear.  Do you mean  “and the mode of reproduction of the plant”?  Or, do you mean the function group to which the species belongs?

Response:Yes, we mean “the mode of reproduction of the plant”. We have revised the sentence.

Line 252, 252. Delete “the” before “mean”

Response:Done.

Line 254. Delete “the” before “total”

Response:Done.

Line 255. Delete “the” before “annual”

Response:Done.

Line 258. Delete “the” before “average”

Response:Done.

 Line 259. Delete “the” before “richness”

Response:Done.

 Line 268.  Not very clear.  I suggest you delete “randomly sampled, then”   

Unless you used a random numbers table to help you select the plants, then you sampled in a haphazard fashion.   Did you also put the seeds in the envelope?  Need to make this clear for the reader.

Response:Many thanks for this valuable suggestion. We have revised the sentence (line 287).

Line 270. Change “part” to “parts”

Response:Done.

Line 271.  Not clear about the weighing.  Did you weight the material as soon as you were back in the lab, i.e. fresh weight?  Or, did you dry the material for several days and then weight it, i.e. air-dried weight?  Or, did you dry the material in a oven to a constant weight, i.e. oven-dry weight?  Need to tell the reader which one is correct.

Response:Thank you very much for your professional advice. We dried the material in a oven to a constant weight. We have added the information in this section (line 289-290).

 Line 275.  Insert “total” before “seed mass”    Is this what you did? 

Response:Done.

Line 276.  Change to “measured six other seed traits, which….”

Response:Done.

Line 293.  Remove the “:” and start a new sentence.    This will solve the run-on sentence problem.

Response:Done.

Reviewer 3 Report

Comments and Suggestions for Authors

Review of the manuscript: Seed Size-Number Trade-Off exists in Graminoids but not in 2 Forbs or Legumes: A study from 11 Common Species in Alpine 3 Steppe Communities

 This manuscript presents a well-executed empirical study exploring the seed size–number trade-off across functional groups in an alpine steppe community. The study is well structured and executed. The introduction is well written and provides a clear and logical background for the study. The research objectives are clearly stated and well structured around three specific questions, and the results are carefully analyzed. While the overall novelty is moderate, the findings contribute valuable empirical data from natural communities. I recommend major revisions to improve clarity in the abstract and discussion, especially to emphasize that the community-level trade-off is mainly driven by graminoids.

Line 108: In the Results section, the authors report a statistically significant negative relationship between seed size and seed number in 2018 (p = 0.0011), yet the R² value is very low (0.092), indicating that only ~9% of the variation in seed number is explained by seed size. While statistical significance is reported, I think this weak correlation is not biologically meaningful.

Line 110: The contrasting results between 2018 and 2019 are interesting, particularly the much stronger relationship observed in 2019 (R² = 0.47) compared to the weak but significant correlation in 2018 (R² = 0.092). I suggest that the authors provide a brief discussion of potential ecological or environmental factors that could explain the difference between years

The authors report a significant seed size–number trade-off at the community level, yet the results show that this relationship is driven solely by graminoids, as it is not significant in forbs or legumes. The community-level pattern is likely an artefact of the strong pattern within graminoids.

The discussion does acknowledge that the seed size–number trade-off was observed only in graminoids, and the authors provide a plausible ecological explanation based on reproductive strategy and resource availability. However, I suggest that this key finding be emphasized more clearly and earlier in the discussion, and that the community-level pattern be interpreted more cautiously, as it seems largely driven by graminoids.

128-131: For clarity and consistency, I suggest reporting the reproductive biomass values in the same functional group order for both years (e.g., always graminoids, forbs, legumes). This would make it easier for readers to follow year-to-year comparisons and reduce cognitive load during interpretation.

Line 140: The conclusion that forbs and legumes are distinct while graminoids overlap with both groups appears to be based solely on visual inspection of the PCA plot. I recommend that the authors report the proportion of variance explained by the first two principal components and include PCA loadings (in a table in the supplementary material), to show how each trait contributes to the ordination.

Figure A4: Please include the loading scores for each trait in the figure caption, so readers can understand which traits drive variation along PC1 and PC2.

Author Response

This manuscript presents a well-executed empirical study exploring the seed size–number trade-off across functional groups in an alpine steppe community. The study is well structured and executed. The introduction is well written and provides a clear and logical background for the study. The research objectives are clearly stated and well structured around three specific questions, and the results are carefully analyzed. While the overall novelty is moderate, the findings contribute valuable empirical data from natural communities. I recommend major revisions to improve clarity in the abstract and discussion, especially to emphasize that the community-level trade-off is mainly driven by graminoids.

Response:Thanks for your valuable comment. Following your comments, we have made the following key revisions to the manuscript: (1) We have rearranged the paragraphs in the Discussion section and modified the expressions in both the Discussion (line187-189, 254) and Abstract (line 23-24, 28) to emphasize that the community-level trade-off is mainly driven by graminoids. (2) We have added several sentences to discuss and explain the changes in R²between 2018 and 2019 (line242-249). (3) We have revised the Results section (line136-138, 151-153) and included additional information on the PCA analysis (Table A2).

Line 108: In the Results section, the authors report a statistically significant negative relationship between seed size and seed number in 2018 (p = 0.0011), yet the R² value is very low (0.092), indicating that only ~9% of the variation in seed number is explained by seed size. While statistical significance is reported, I think this weak correlation is not biologically meaningful.

Response:Thanks for your valuable comment. You are right, the R²value is low in 2018. But we think the low R²also have biological meaning. This results can be attributed to the multitude of potential factors influencing seed number, of which seed size was the only factor considered in our current analysis. Indeed, we contend that a low R²value is often observed in studies involving field sampling. As an illustrative example, a study published in Science reported an R²of merely 0.03, yet this was considered acceptable within its context.

Reference

Langhammer, P. F. et al. The positive impact of conservation action. Science 384, 453-458, doi:doi:10.1126/science.adj6598 (2024).

Line 110: The contrasting results between 2018 and 2019 are interesting, particularly the much stronger relationship observed in 2019 (R² = 0.47) compared to the weak but significant correlation in 2018 (R² = 0.092). I suggest that the authors provide a brief discussion of potential ecological or environmental factors that could explain the difference between years.

Response:Thanks for your valuable suggestion. As your suggestion, we conducted a comparative analysis of precipitation and reproductive allocation data for 2018 and 2019. Our results indicated that precipitation in 2019 was 14.67% higher than in 2018, while the temperature decreased by 3.31%. In alpine steppe ecosystems, precipitation is a crucial limiting factor. Under conditions of higher precipitation, graminoids may exhibit a greater propensity for asexual reproduction, consequently allocating fewer resources to sexual reproduction (the reproductive allocation is 16.26% and 9.98% in 2018 and 2019, respectively. Figure 3) (Zhu et al, 2007). This constraint on reproductive resources led to a more pronounced trade-off between seed size and reproduction (Figure 2). We also added a brief discussion in line xxx.

Reference

Zhu, Y., Ala, T., Dong, M., and Huang, Z. (2007). Effects of increasing water or nutrient supplies on reproduction trade-offs in the natural populations of clonal plant Hedysarum leave. J. Plant. Ecol. 31, 658–664. doi: 10.17521/cjpe.2007.0085

The authors report a significant seed size–number trade-off at the community level, yet the results show that this relationship is driven solely by graminoids, as it is not significant in forbs or legumes. The community-level pattern is likely an artefact of the strong pattern within graminoids.

The discussion does acknowledge that the seed size–number trade-off was observed only in graminoids, and the authors provide a plausible ecological explanation based on reproductive strategy and resource availability. However, I suggest that this key finding be emphasized more clearly and earlier in the discussion, and that the community-level pattern be interpreted more cautiously, as it seems largely driven by graminoids.

Response:Many thanks for your valuable suggestion. We have rearranged the order of the original paragraphs in the Discussion to emphasize the significant contribution of graminoid species to the community-level seed size-number trade-off. Additionally, we have made several modifications to the text to further highlight the unique role of graminoids (line187-189, 254). At the same time, we have revised the abstract to reflect these changes (line 23-24, 28).

128-131: For clarity and consistency, I suggest reporting the reproductive biomass values in the same functional group order for both years (e.g., always graminoids, forbs, legumes). This would make it easier for readers to follow year-to-year comparisons and reduce cognitive load during interpretation.

Response:Many thanks for your valuable suggestion. We have revised the sentence as your suggestion (line 137-138).

Line 140: The conclusion that forbs and legumes are distinct while graminoids overlap with both groups appears to be based solely on visual inspection of the PCA plot. I recommend that the authors report the proportion of variance explained by the first two principal components and include PCA loadings (in a table in the supplementary material), to show how each trait contributes to the ordination.

Response:Many thanks for this valuable suggestion. We have reported the proportion of variance explained by the first two principal components in Results section (line 151-153) and supplied Table A2 which included PCA loadings for each seed trait.  

Figure A4: Please include the loading scores for each trait in the figure caption, so readers can understand which traits drive variation along PC1 and PC2.

Response:Many thanks for this valuable suggestion. We have provided the PCA loadings in Table A2; accordingly, we have added a note in the figure caption.

Round 2

Reviewer 3 Report

Comments and Suggestions for Authors

Review of the manuscript: Seed Size-Number Trade-Off exists in Graminoids but not in 2 Forbs or Legumes: A study from 11 Common Species in Alpine 3 Steppe Communities

I have carefully reviewed the revised manuscript and the authors’ detailed responses to my previous comments. The manuscript has been substantially improved. While the novelty of the findings is moderate, the study provides valuable empirical data from a natural alpine steppe community. I have no major concerns with the manuscript except regarding the PCA.

Although the authors have now reported the variance explained and provided PCA loadings (Table A2), the interpretation of the PCA remains minimal. The current text (lines 146–148) simply states that forbs and legumes are distinct while graminoids overlap, without further biological explanation. I encourage the authors to briefly interpret the axes in ecological terms. For example, PC1 clearly reflects the seed size–number trade-off, with seed number loading positively and seed size-related traits (size, width, height) loading negatively. Including a brief interpretation of PC2 would help readers understand additional functional differences among species beyond the seed size–number trade-off captured by PC1. This would make the PCA more informative and directly link it to the theoretical framework of the study.

It is not sufficient to present only the species points (functional group averages). As currently shown, the figure does not allow readers to understand which seed traits drive the observed patterns. My initial suggestion was to include the variable loadings (arrows) in the PCA biplot, which is the standard way to visualize trait contributions along PC1 and PC2. Plotting the loadings together with the species scores would make the ecological interpretation much clearer.

Author Response

I have carefully reviewed the revised manuscript and the authors’ detailed responses to my previous comments. The manuscript has been substantially improved. While the novelty of the findings is moderate, the study provides valuable empirical data from a natural alpine steppe community. I have no major concerns with the manuscript except regarding the PCA.

Although the authors have now reported the variance explained and provided PCA loadings (Table A2), the interpretation of the PCA remains minimal. The current text (lines 146–148) simply states that forbs and legumes are distinct while graminoids overlap, without further biological explanation. I encourage the authors to briefly interpret the axes in ecological terms. For example, PC1 clearly reflects the seed size–number trade-off, with seed number loading positively and seed size-related traits (size, width, height) loading negatively. Including a brief interpretation of PC2 would help readers understand additional functional differences among species beyond the seed size–number trade-off captured by PC1. This would make the PCA more informative and directly link it to the theoretical framework of the study.

It is not sufficient to present only the species points (functional group averages). As currently shown, the figure does not allow readers to understand which seed traits drive the observed patterns. My initial suggestion was to include the variable loadings (arrows) in the PCA biplot, which is the standard way to visualize trait contributions along PC1 and PC2. Plotting the loadings together with the species scores would make the ecological interpretation much clearer.

Response:Thank you for your valuable comment. We regret that we did not fully comprehend your initial suggestion. In accordance with your comment, we have now revised Figure A3 to include the variable loadings (arrows). We highly appreciate your insightful interpretation regarding the ecological significance represented by the PC1 axis. Concerning the PC2 axis, we propose that it represents the capacity of seeds to persist within the soil seed bank. This interpretation is based on the premise that small and spherical seeds are more likely to be buried deeply and thus persist for longer durations in the soil compared to large and nonspherical seeds (Wang et al. 2024). Furthermore, the high seed carbon content additionally contributes to enhanced seed longevity and survival in soil (Long et al. 2014). We have supplemented this information into the Results (line 151-155) and Discussion (line 215-218) section.

Reference

Long, R. L., M. J. Gorecki, M. Renton, J. K. Scott, L. Colville, D. E. Goggin, L. E. Commander, D. A. Westcott, H. Cherry, and W. E. Finch‐Savage. 2014. The ecophysiology of seed persistence: a mechanistic view of the journey to germination or demise. Biological Reviews 90:31-59.

Wang, X., W. Ge, M. Zhang, E. Fernández‐Pascual, A. Moles, A. Saatkamp, S. Rosbakh, H. Bu, P. Panahi, and M. Ma. 2024. Large and non-spherical seeds are less likely to form a persistent soil seed bank. Proceedings of the Royal Society B: Biological Sciences 291.